# Maternal Autogenous Inactivated Virus Vaccination Boosts Immunity to PRRSV in Piglets

**DOI:** 10.3390/vaccines9020106

**Published:** 2021-01-31

**Authors:** Andrew R. Kick, Zoe C. Wolfe, Amanda F. Amaral, Lizette M. Cortes, Glen W. Almond, Elisa Crisci, Phillip C. Gauger, Jeremy Pittman, Tobias Käser

**Affiliations:** 1Department of Population Health and Pathobiology, College of Veterinary Medicine, North Carolina State University, Raleigh, NC 27607, USA; andrew.kick@westpoint.edu (A.R.K.); zcwolfe@ncsu.edu (Z.C.W.); afamaral@ncsu.edu (A.F.A.); lmlorenz@ncsu.edu (L.M.C.); gwalmond@ncsu.edu (G.W.A.); ecrisci@ncsu.edu (E.C.); 2Department of Chemistry & Life Science, United States Military Academy, West Point, NY 10996, USA; 3Veterinary Diagnostic and Production Animal Medicine, Iowa State University, Ames, IA 50011, USA; pcgauger@iastate.edu; 4Smithfield Foods, Smithfield, VA 23430, USA; jpittman@smithfield.com

**Keywords:** maternal vaccination, autogenous inactivated vaccine, transfer of immunity, humoral immune response, cell-mediated immune response, T cells, PRRSV, swine, IFN-γ producing B cells, CD4 T_EMRA_ cells

## Abstract

Maternal-derived immunity is a critical component for the survival and success of offspring in pigs to protect from circulating pathogens such as Type 2 Porcine Reproductive and Respiratory Syndrome Virus (PRRSV-2). The purpose of this study is to investigate the transfer of anti-PRRSV immunity to piglets from gilts that received modified-live virus (MLV) alone (treatment (TRT) 0), or in combination with one of two autogenous inactivated vaccines (AIVs, TRT 1+2). Piglets from these gilts were challenged with the autogenous PRRSV-2 strain at two weeks of age and their adaptive immune response (IR) was evaluated until 4 weeks post inoculation (wpi). The systemic humoral and cellular IR was analyzed in the pre-farrow gilts, and in piglets, pre-inoculation, and at 2 and 4 wpi. Both AIVs partially protected the piglets with reduced lung pathology and increased weight gain; TRT 1 also lowered piglet viremia, best explained by the AIV-induced production of neutralizing antibodies in gilts and their transfer to the piglets. In piglets, pre-inoculation, the main systemic IFN-γ producers were CD21α^+^ B cells. From 0 to 4 wpi, the role of these B cells declined and CD4 T cells became the primary systemic IFN-γ producers. In the lungs, CD8 T cells were the primary and CD4 T cells were the secondary IFN-γ producers, including a novel subset of porcine CD8α^−^CCR7^−^ CD4 T cells, potentially terminally differentiated CD4 T_EMRA_ cells. In summary, this study demonstrates that maternal AIV vaccination can improve protection of pre-weaning piglets against PRRSV-2; it shows the importance of transferring neutralizing antibodies to piglets, and it introduces two novel immune cell subsets in pigs—IFN-γ producing CD21α^+^ B cells and CD8α^−^CCR7^−^ CD4 T cells.

## 1. Introduction

Maternal-derived immunity (MDI) in mammals is an awe-inspiring symphony (“harmonious complexity”—[1]). In this system, or symphony, a mother transfers immune protection (temporary and/or enduring) to her offspring either during gestation through the placenta, after birth through mammary secretions, or both [2]: humans receive immunoglobulins during gestation and eventually through mammary secretions [3]; dogs and rodents receive immunoglobulins during gestation and through mammary secretions, and pigs and other livestock only receive immunoglobulins through mammary secretions. Similar to other species, piglets receive this MDI in the form of immunoglobulins, immune cells, and other immune-related molecules [4,5]. While immunoglobulins can be received from different sows (= cross-fostering), only biological mothers can successfully transfer immune cells to their offspring [5], and the quantity of colostrum intake in piglets significantly decreases mortality and improves performance [6]. These immune cells include B cells [7], as well as different T-cell subsets—CD4, CD8 and mainly T-cell receptor (TCR)-γδ T cells [8]. These T-cell subsets can then be further distinguished based on their differentiation stage and homing pattern. In pigs, the CD8α chain is used to differentiate CD8α^‑^ naïve from CD8α^+^ antigen-experienced or memory CD4 T cells [9,10,11]. The expression of the chemokine receptor (CCR) 7 further distinguishes CCR7^+^ lymph node homing from CCR7^‑^ peripheral tissue-homing T cells [12]. With a combined CD8α/CCR7 analysis, we can then distinguish multiple different T-cell subsets: CD8α^−^CCR7^+^ naïve CD4 T cells, CD8α^+^CCR7^+^ central memory (T_CM_) CD4 T cells, and CD8α^+^CCR7^−^ effector memory (T_EM_) CD4 T cells; peripheral tissue-homing CCR7^−^ and lymph node homing CCR7^+^ CD8 T cells, and CD8α^−^CCR7^−^, CD8α^+^CCR7^−^, and CD8α^+^CCR7^+^ TCR-γδ T cells. While we previously reported that porcine TCR-γδ T cells possess an inverse homing pattern compared to their TCR-αβ counterparts, the specific role of these different subsets has yet to be described [13]. Both, immune cell subsets and antibodies are transferred to the offspring to protect them against various pathogens [4,5].

For decades, the most disastrous pathogen for pigs has been the Porcine Reproductive and Respiratory Syndrome Viruses (PRRSV, [14,15]). Two main types of PRRSV exist—type 1 and 2 (PRRSV-1 and 2); their high diversity and immunosuppressive capacities make PRRSV a problematic disease, leading to tremendous economic losses for the swine industry [16] and so far, while vaccines provide partial protection against PRRSV, they have not been able to fully control this disease. 

Two main vaccine types are available to protect against PRRSV—modified live virus (MLV) vaccines and inactivated vaccines, mainly autogenous inactivated vaccines (AIVs); however, both types face their own challenges. The long time to develop the very immunogenic MLV vaccines means they are usually more heterologous to farm-prevalent PRRSV strains, and while AIVs are directed against these farm-prevalent strains, their inactivated nature makes them generally less immunogenic. These challenges lead to a lack of protection especially in young pigs that can be attributed to three factors: first, the high mutation rate of PRRSV limits heterologous protection, so it reduces MLV vaccine efficacy; second, its immunosuppressive capacity limits weaker vaccines (e.g., AIVs) and its delay of the induction of a protective immune response leaves young pigs especially susceptible [17]; and third, our understanding of the immune response to PRRSV, especially in piglets, is limited. Loving and Lunney produced a nice review regarding the general understanding of the immune response to PRRSV [18,19]. 

Mainly based on the extended time to establish vaccine-induced immunity to PRRSV, swine producers often vaccinate gilts and sows to enable transfer of immunity to piglets. Research on maternal vaccination using a commercial MLV against type PRRSV-1 did not impede vaccination of 2 or 3 week old piglets and it partially protected piglets against heterologous challenge [20]. This vaccination strategy also induced PRRSV-1-specific IgG, IgA, and interferon (IFN)-γ secreting cells in the colostrum and milk of vaccinated sows. These IFN-γ producing cells are highly important in anti-viral responses, including in PRRSV [19,21]. This vaccination also induced serum anti-PRRSV IgG, which was detectable after six weeks of age for unvaccinated piglets and remained stable in vaccinated piglets prior to challenge; yet, IFN-γ secreting cells could not be distinguished from background levels in pre-weaned piglets [22]. While the use of MLVs is an industry standard, a combined vaccination regimen of an MLV prime and AIV boost vaccination seems promising to overcome each of the downsides of independently using MLVs and AIVs. 

Therefore, the main goal of this study was to determine not only if, but also how, maternal vaccination with two different AIVs (treatment (TRT) 1+2) can boost an industry standard MLV vaccination alone (TRT 0), to better protect the offspring from early PRRSV-2 challenge. To this end, we studied the vaccine-induced humoral and cell-mediated immune (CMI) response in these gilts and we challenged early weaners from gilts within each of these groups at 2 weeks of age. The systemic and local immune response of these piglets was followed for 2–4 weeks. We show that AIV boost vaccination could improve the protection of piglets against PRRSV challenge at 2 weeks of age and we also identify the underlying immune mechanisms and characterize the anti-PRRSV humoral and CMI response in these young pigs. In addition, we detected two new cell types in pigs: (1) CD8α^−^CCR7^‑^ CD4 T cells that strongly contribute to the IFN-γ production in bronchoalveolar lavage (BAL)—potential CD4 T_EMRA_ cells, and (2) a major source of the anti-PRRSV response especially in piglets—IFN-γ producing B cells.

## 2. Materials and Methods 

### 2.1. Study Design, Animals, and Sample Processing

The purpose of this study was to evaluate the effect of different maternal PRRSV-2 vaccination series on protective immunity in weaned piglets against PRRSV-2. Prior to the study, a circulating PRRSV-2 strain (restriction fragment length polymorphism (RFLP), 1-7-4) from a North Carolina pork company was isolated and propagated by two independent biological sciences companies; they produced two distinct autogenous vaccines. Both vaccines contained different proprietary components but contained the same inactivated 1-7-4 PRRSV-2 strain. The design of this study is illustrated in Figure 1. 

Both vaccinations were administered intramuscularly in a series of four doses prior to farrowing: in accordance with the North Carolina pork company corporate vaccination schedule, 46 PRRSV-2 negative gilts in a commercial sow farm were vaccinated with the Ingelvac PRRS MLV vaccine (Boehringer Ingelheim Vetmedica GmbH, Germany) nine weeks prior to breeding. To maintain consistent terminology throughout this study, the enrolled sexually mature female pigs (here defined as gilts) will be referred to as gilts even after they were bred and gave birth (commonly referred to as sows). Four weeks prior to breeding, blood samples were collected from all gilts for serum and then gilts were allotted into three treatments: TRT 0 (MLV vaccination only); TRT 1 (MLV vaccination + 4 doses of autogenous (1-7-4) inactivated vaccine 1); or TRT 2 (MLV vaccination + 4 doses of autogenous (1-7-4) inactivated vaccine 2). Gilts that did not breed were dropped from the study. At 3 weeks before farrowing, blood was collected from pregnant gilts for the isolation of serum and peripheral blood mononuclear cells (PBMC). At this point, five gilts remained in both TRT 0 and TRT 2; nine gilts remained in TRT 1. Within these gilts, we selected three gilts per group to provide the piglets for the challenge study. The selected gilts had an adequate litter size (>7) and the highest serum fluorescent focus neutralization (FFN) titer (primary immune criteria) and the highest CD4 T-cell IFN-γ response (secondary immune criteria) against the AIV challenge strain within their group. For TRT 1+2, these FFN titers ranged from 1:4 to 1:128. Since TRT 0 did not have any titers against the AIV strain, we chose the gilts with the highest anti-MLV titers; these titers ranged from 1:16 to 1:32. The IFN-γ response within all selected gilts ranged from 0.09–0.21% of the total CD4 T cells. Each of these gilts provided four piglets for the challenge study. At 2 weeks of age (= 0 weeks post infection, wpi), the four piglets with the most average litter weights were selected for the challenge study and transported to the Biosecurity Level 2 housing facilities of the North Carolina State University (NCSU) Laboratory Animal Resources (LAR) (Raleigh, North Carolina). Here, they were randomly assigned to one of six pens in the same room; two pigs per treatment were housed in each pen (one per gilt). In summary, there were a total of 36 piglets in the study with 12 piglets per treatment group. During the study, four piglets died prior to the end of the study: three piglets from one gilt (TRT 0) died within 30 h of arrival at LAR and were dropped from the study. Additionally, one pig (TRT 2) was euthanized between week three and four of the study due to a high fever (>106° F) and lack of response to physical stimulation. This pig was included in all analyses through week 3 and its lungs were assessed for pathology. No flow cytometry (FCM) analysis occurred for this animal at necropsy.

One day after weaning and arrival at NCSU, all piglets were inoculated intranasally with 1 × 10^5.5^ PRRSV-2 viral particles (autogenous strain 1-7-4) in 1 ml 1X phosphate buffered saline (PBS, Corning, Corning, NY, USA) (500 μL/nostril), maintained in a supine position for 30 seconds and returned to their pens. Half the animals were sacrificed at 2 wpi and the remaining animals were sacrificed at 4 wpi. At necropsy, BAL, lungs, and tracheobronchial lymph nodes (TrBr LNs) were collected and processed as previously described [13]. 

Sampling occurred prior to inoculation (0 wpi) and then weekly, as shown in Figure 1. Serum was collected in Serum Separator Tubes (SST) tubes (Becton Dickinson (BD) Biosciences, San Jose, CA, USA) and blood for PBMC isolation was collected in heparin tubes (BD Biosciences). Animal housing was in accordance with the National Institutes of Health guide for research animals. Piglets had unlimited access to water, were fed a transitional weaning diet, then regular diet twice daily, and maintained at 83 °F with a 12-hour light/dark cycle. The experimental procedures were approved by the NC State University Institutional Animal Care and Use Committee (IACUC) ID# 17-166A (29 November 2017).

### 2.2. Virus Propagation, Titration, and Evaluation

The PRRSV-2 viral isolate (strain, 1-7-4) was propagated on PRRSV-negative porcine alveolar macrophages (PAMs) in RPMI-1640, 1× with L-glutamine (Corning, Corning, NY, USA) supplemented with 10% fetal bovine serum (FBS) (VWR, Radnor, PA, USA) and 1X antibiotic-antimycotic (Corning) (referred to as RPMI-complete) for approximately 30 h in T-75 flasks (Sarstedt, Nümbrecht, Germany) until a cytopathic effect was observed. Flasks were freeze-thawed one time and the supernatant was spun at 2200 *g* at 4 °C for 10 min. To concentrate the virus, the supernatant was then transferred to 36 mL Nalgene centrifuge tubes (Thermo Fisher Scientific, Rochester, NY, USA) and spun in a Sorvall 100S Ultracentrifuge (Sorvall, Thermo Fisher Scientific), Newtown, CT, USA) at 73,000 *g* at 4 °C for 2 h. The supernatant was discarded, and the pellet was resuspended in media (RPMI complete) and stored in 100 µL aliquots at −80 °C. The tissue culture infectious dose 50 (TCID_50_) titers of viral stock solutions were determined utilizing PAMs as previously described (Spearman–Karber TCID50 method according to the OIE manual of diagnostic tests [World Organisation for Animal Health, formerly the Office International des Epizooties (OIE), “Chapter 2.8.7 Porcine Reproductive and Respiratory Syndrome”, Terr. Man., No. May 2015, 2015].

### 2.3. Viremia

Isolated serum was analyzed by the North Carolina pork company’s diagnostic laboratory using Reverse-Transcription quantitative polymerase chain reaction (RT qPCR) with a 5X MagMAX pathogen RNA/DNA kit (Thermo Fisher Scientific) and VetMAX PRRSV qPCR kit (Thermo Fisher Scientific). Vaccinated gilt serum was PRRSV-negative (qPCR) at farrowing as determined by the Iowa State University Veterinary Diagnostic Laboratory (ISU VDL, Ames, IA, USA). The ΔCt values were calculated by subtracting the *Ct* value of a sample from the *Ct* value considered as PRRSV-negative (Ct_neg_ = 37) for PRRSV: ΔCt = 37 − Ct_x_.

### 2.4. Neutralizing Antibody Validation and Determination of Positive Samples

Serum samples from gilts and piglets were shipped to South Dakota State University Animal Research and Diagnostic Laboratory (Brookings, SD) who completed PRRSV-2 neutralizing antibody (NA) determination utilizing FFN and the 1-7-4 inoculation strain. A positive result is the highest serum dilution with a 90% or higher reduction in the number of fluorescent focus forming units [23].

### 2.5. Necropsy Procedures, Scoring, and Tissue Cell Iisolations

At necropsy (2 or 4 wpi), animals were sacrificed using lethal intravenous injection of Euthasol (Covetrus, MW, 390 mg/mL pentobarbital, 50 mg phenytoin/mL). Then, the lungs, thymus, and TrBr LNs were harvested. For the lungs, dorsal and ventral photos were taken; lungs were flushed with 1× PBS to collect the bronchial alveolar lavage (BAL) and five total samples were taken for histology: one each from the right and left caudal lobes, right and left middle lobes and then one from the accessory lobe. Histology samples were fixed in Formaldehyde/Zn fixative (Electron Microscopy Sciences, Hatfield, PA) for 24 h, transferred to 70% ethanol and stained with hematoxylin and eosin (H&E). Stained slides (histology) and lung photos (gross pathology) were shipped to ISU VDL and scored by a veterinary diagnostic pathologist who was blinded to treatments and necropsy dates; lung lesion and interstitial pneumonia scoring was performed as previously described [24]. Dorsal and ventral views of lung photos were evaluated for the percentage of the lung affected with pneumonia. Specifically, the percentage of pneumonia was estimated visually for each lung lobe, and the total percentage for the entire lung was calculated based on the weighted proportion of each lobe relative to the total lung volume. Each of the four cranioventral lobes represented 10%, the accessory lobe 5%, and each of the two caudal lobes represented 27.5% of the total lung. Lung histopathology sections were blindly examined for microscopic lesions and each lobe was given an estimated score based on the severity of the interstitial pneumonia as follows: 0 = no microscopic lesions; 1 = minimal interstitial pneumonia; 2 = mild interstitial pneumonia; 3 = moderate interstitial pneumonia; 4 = abundant interstitial pneumonia; 5 = severe, multifocal interstitial pneumonia; 6 = severe, diffuse interstitial pneumonia. For all collected tissues (lungs, thymus, and TrBr LNs), the tissue was cut into small pieces and pressed through stainless steel round drain strainers (Grainger, Lake Forest, IL, USA) into 50 ml tubes with ice-cold PBS to elute single cell suspensions. Fibrous clumps were filtered with 40 μm filters (BD Biosciences, San Jose, CA, USA).

### 2.6. Peripheral Blood Isolation, Viral Stimulation, and IFN-γ FCM Staining/Analysis

PBMC isolation was performed using Ficoll-Paque density centrifugation (GE Healthcare, Uppsala, Sweden) and SepMate-50 ml tubes (STEMCELL Technologies, Vancouver, BC, Canada) in accordance with the manufacturer’s protocol. PBMC and single cells suspension cell counts were completed on a CASY cell counting system (Schärfe System, now: Roche Innovatis AG, Bielefeld, Germany). For the IFN-γ production assay, 5 × 10^5^ PBMCs and lymphocytes from lungs, and TrBr LNs were plated in eight replicates in a 96-well round-bottom plate (Corning) in 100 μl RPMI-complete and rested overnight in an incubator at 37° C and 5% CO_2_. The next morning 100 μL of RPMI-complete containing PRRSV strain 1-7-4 (Multiplicity of Infection, MOI = 0.1) was added to each well; plates were returned to the incubator for 24 h of viral stimulation. For the last 4 h of culture, monensin (5 µg/mL, Alfa Aesar, Haverhill, MA, USA) was added to block IFN-γ release from the cells. Upon completion of the 24 h incubation, the replicates were pooled and stained as stated in Table 1; on average 276,756 single living lymphocytes (SLLs) were recorded on a Cytoflex using the CytExpert software (Beckman Coulter, Brea, CA, USA). Data analysis was performed with FlowJo version 10.6.1 (FLOWJO LLC). Gating hierarchy for the analysis of the IFN-γ response is shown in Figures 4 and 5 for PBMC and in Appendix A and Figure 7 for BAL, lung tissue, and TrBr LNs.

### 2.7. Statistical Analysis

Statistical analyses were performed using Graphpad Prism 8 (Graphpad Software, San Diego, CA, USA). Data with a full data set were analyzed by either two-way (throughout the study) or one-way (at necropsy) ANOVAs. Data with missing data points (mainly due to necropsy of 50% of the pigs at 2 wpi) were analyzed by a mixed-effects model (Restricted maximum likelihood, REML). All multiple comparisons were performed with Dunnett’s multiple comparison test. Data illustration was performed using violin plots showing the distribution, individual points, median (as applicable, dashed black line), and 25th/75th intervals (as applicable, colored line). Correlation analyses between gilt and piglet nAb levels were performed using Pearson correlation coefficients and a two-tailed 95% confidence interval.

## 3. Results

### 3.1. Study Design and Vaccine Efficacy—Clinical Signs, Viremia, Weight Gain, and Lung Pathology

The study was conducted as depicted in Figure 1. Upon PRRSV inoculation, pigs displayed mild PRRSV symptoms beginning within 3–4 days of infection—fever [mainly at 1 wpi, d.n.s.], lethargy, reduced food consumption, mild coughing/sneezing. No significant differences were observed for these symptoms and they persisted to varying degrees until necropsy. Figure 2 illustrates viremia, weekly body weight gains as well as gross and histopathology scores. 

As shown in Figure 2A, all pigs were PRRSV-2 negative at weaning and intranasal inoculation with PRRSV-2 induced a viremia that peaked at 2 wpi. While there was no difference in viremia between TRT 0 and TRT 2, TRT 1 had significantly lower viremia in the first two weeks than the control group TRT 0. The weight gains of pigs in all groups were very similar for the first three weeks; yet, at the end of study—and compared to TRT 0—TRT 2 and TRT 1 had significantly higher weekly weight gains—Figure 2B. Most notably, while lung pathology is generally resolved in PRRSV infection and was therefore similar between groups at 4 wpi (d.n.s.), both maternal AIV vaccinations, so TRT 1+2, significantly reduced the lung pathology at 2 wpi: both decreased the percentage levels of macroscopic lesions and/or interstitial pneumonia (Figure 2C,D). Prevalence of macroscopic lung lesions in lungs from TRT 0 was between 20–70%; in contrast, most pigs in TRT 1+2 had a lung lesion prevalence below 20%—Figure 2C. Similarly, while the median histopathology score for interstitial pneumonia in lungs from TRT 0 pigs was three, it was 1.8 for TRT 1 and 1.2 for TRT 2, respectively. Compared to TRT 0, this demonstrates a tendency towards significance (*p* = 0.07) and a significant decrease in lung histopathology in pigs from AIV vaccinated gilts—Figure 2D. These data demonstrate that AIV vaccination can significantly reduce viremia and lung pathology, additionally, they also show that maternal AIV vaccination against PRRSV-2 can boost the immunity of 2-week old piglets.

### 3.2. Induction of Maternal Neutralizing Antibodies and Their Transfer to Piglets

The induction of a humoral immune response in gilts and their offspring was analyzed by determining NA FFN titers (Figure 3).

Vaccination of all groups with the MLV induced NA titers in gilts of between four and 64 with no significant differences between the three groups (d.n.s.). In contrast, while MLV vaccination alone (TRT 0) did not include NAs against the heterologous 1-7-4 strain, both AIV boost vaccinations induced NA titers of up to 1:16 (for TRT 2) and even 1:128 (for TRT 1)—Figure 3A. These serum NAs were maternally transferred to piglets (Figure 3B): While TRT 0 piglets showed no NA titers at 0 wpi, TRT 1 and TRT 2 piglets possessed serum NAs to strain 1-7-4 prior to infection which directly correlated with maternal NA levels (Figure 3C, r = 0.886, *p* < 0.0001). These NA titers decreased over the first two weeks, before stabilizing by 4 wpi. Yet, compared to TRT 0, piglets from AIV-vaccinated gilts (TRT 1+2) had significantly higher serum NA titers already before challenge and these increased titers lasted until 2–3 wpi, so until 4–5 weeks of age—Figure 3B.

### 3.3. The Systemic Cell-Mediated Immune Response to PRRSV

The systemic cell-mediated immune response to the 1-7-4 PRRSV strain was analyzed by studying the IFN-γ production of lymphocytes and the determination of the contributing immune cell subsets using multi-color flow cytometry (Figure 4). 

Figure 4A shows the gating hierarchy. Figure 4B shows the IFN-γ production in single living lymphocytes (SLLs) and the contribution of B cells, NK cells, CD4, CD8 and TCR-γδ T cells to this production. At all time points, the systemic IFN-γ response in SLLs was low with median frequencies ranging from ~0.02% (mainly at 4 wpi) to 0.12% in gilts. Neither NK cells nor TCR-γδ T cells were responsible for more than 10% of the produced IFN-γ; consequently, B cells, and CD4 and CD8 T cells were the main contributors to the SLL IFN-γ production.

In gilts, the vast majority of this IFN-γ was produced by CD4 T cells. Nevertheless, analysis of the IFN-γ in piglets before PRRSV inoculation showed that B cells were the main contributors to the systemic IFN-γ production. At this time point, we also recorded the only significant increase in the percentage of IFN-γ producing SLLs by one of the AIV vaccinations—TRT 2. While the contribution of immune cells to this increased production did not reveal differences, IFN-γ production within two immune cell subsets was significantly increased in this TRT 2 group as well—B cells and TCR-γδ T cells (Appendix A). Over the time course of PRRSV infection, the overall SLL IFN-γ response slightly declined and the contributions of the different immune cell subsets shifted notably. The contribution of B cells dropped by more than half from over 40% to under 20%, and the contribution of CD8 T cells increased from below 5% to ~ 20%. Yet, while CD4 T cells produced ~20% of the total IFN-γ at 0 and 2 wpi, with a contribution of over 40% they were the strongest IFN-γ producing immune cell subset at 4 wpi. 

### 3.4. The Differentiation and Homing of IFN-γ Producing T-Cell Subsets in Blood

Differentiation and homing of immune cells is critical for the vaccine-induced immunity. Differentiation into different memory cells can either speed up their development of an effector function (for T_CM_s) or provide them with an immediate effector function (for T_EM_s). Location is also critical to ensure a fast and directed response to a pathogen response. While T_CM_s home to secondary lymphoid tissues where they can undergo rapid proliferation, T_EM_s home to peripheral tissues to exert their effector function. Based on this relevance, we studied the differentiation and homing patterns of the IFN-γ producing CD4, CD8 and TCR-γδ T-cell subsets. These patterns are shown in Figure 5.

In gilts, over 80% of the systemic IFN-γ producing CD4 T cells belong to the CD8α^+^ memory subsets—T_CM_s produce over 50% and T_EM_s produce 20–30%. In contrast, in pre-inoculated piglets at 0 wpi, the vast majority of IFN-γ within CD4 T cells is produced by naïve T cells—~80%. Upon challenge, the role of naïve CD4 T cells decreases and memory T cells start to increase—at 2 wpi T_CM_s, and then also at 4 wpi T_EM_s (Figure 5B, “CD4^+^ T cells” panels). In gilt CD8 T cells, 60 and 40% of IFN-γ is produced by CCR7^+^ and CCR7^−^ cells, respectively. In piglets, homing of IFN-γ CD8 T cells was harder to assess based on their low frequency of IFN-γ production. This is also reflected in the higher variance of these frequencies. In general, pre and 2 wpi, IFN-γ was mainly produced by CCR7^+^ cells; in contrast, at 4 wpi, CCR7^−^ CD8 T cells were the main IFN-γ producers (Figure 5B, “CD8 T cells” panels). 

As described in the literature, porcine TCR-γδ T cells possess a homing pattern that is inverse to their CD4 and CD8 TCR-αβ counterparts; while naïve TCR-αβ T cells express CCR7 and home to secondary lymphoid tissues, naïve TCR-γδ T cells do not express CCR7 and home to peripheral tissues [13]. In gilts, ~20–50% and ~30–60% of systemic IFN-γ was produced by naïve CCR7^−^CD8α^−^ cells and CCR7^−^CD8α^+^ tissue-homing antigen experienced TCR-γδ T cells, respectively; only a minority (0–10%) was produced by CCR7^+^CD8α^+^ lymph node homing, antigen experienced TCR-γδ T cells (Figure 5B, “TCR-γδ T cells” panels). As for CD8 T cells, the low frequency of IFN-γ producing TCR-γδ T cells in piglets led to high variability in their differentiation and homing patterns. Yet, one general conclusion can be drawn: conversely to gilts, IFN-γ in piglet TCR-γδ T cells was mainly produced by naïve CCR7^−^CD8α^−^ TCR-γδ T cells.

### 3.5. The Local Cell-Mediated Immune Response to PRRSV—BAL, Lung, and Tracheobronchial Lymph Nodes

While the systemic CMI response was studied in gilts and piglets throughout the challenge study, the local CMI response was studied at necropsy at 2 and 4 wpi, as shown in Figure 6A,B.

At 2 wpi, no differences between groups were observed for the overall IFN-γ production within SLLs; this production was 0.1–0.5% lower in the TrBr LNs than in the lung and BAL (each ~0.2–2%). For the TrBr LNs, B cells were the major producer of IFN-γ at this time point. B cells from TRT 2 showed a higher contribution to this IFN-γ production by number in TrBr LNs and significantly in lung tissue. Yet, CD8 T cells were the major contributor to the IFN-γ production in the lung and they share this spot with CD4 T cells in the BAL.

At 4 wpi, IFN-γ production in TrBr LNs was very low. B cells were still the main contributors but CD4 and CD8 T cells also provided ~20% each of the produced IFN-γ amount. In lung tissue and BAL, CD8 T cells could expand their role in IFN-γ production by producing 20–80% of the overall IFN-γ content. While B cells and TCR-γδ T cells showed almost no IFN-γ production, NK cells and CD4 T cells also produced up to ~10 and 25% of the IFN- γ, respectively.

These data show that B cells are the main producers of IFN-γ in TrBr LNs and that CD8 T cells are the main IFN-γ producers in lung tissue and BAL.

### 3.6. The Differentiation and Homing of IFN-γ Producing T-Cell Subsets in BAL, Lung, and Tracheobronchial Lymph Nodes

The differentiation and homing patterns of CD4, CD8, and TCR-γδ T cell subsets were systemically analyzed by their CD8α and CCR7 expression (Figure 7).

While the low frequencies of IFN-γ^+^ cells limit the analysis of the differentiation of TCR-γδ T cells, and also limit the analysis of CD4 T cells at 2 wpi, these analyses provided a valuable insight into the differentiation of CD4 T cells at 4 wpi and of CD8 T cells at both 2 and 4 wpi. As expected, the majority these cells in TrBr LNs express the lymphoid tissue homing marker CCR7. In contrast, both the majority of IFN-γ^+^ CD4 and CD8 T cells are CCR7^−^ in the lungs and BAL. 

At 4 wpi, IFN-γ^+^ CD4 T cells show a remarkable difference in their CD8α and CCR7 expression, even at 4 wpi, IFN-γ^+^ CD4 T cells in TrBr LNs mainly belong to CD8α^−^CCR7^+^ naïve T cells. In strong contrast, about half of the IFN-γ^+^ CD4 T cells in lungs and BAL are CD8α^+^CCR7^−^ T_EM_ cells, while even more remarkable, is that 25% (lung) and 42% (BAL) of these cells IFN-γ^+^ CD4 T cells are CD8α^−^CCR7^−^—an immune phenotype that has not yet been described in pigs.

## 4. Discussion

The purpose of the present study was to answer the questions (i) if and (ii) how AIV vaccinations (TRT 1+2) can boost an industry standard gilt MLV vaccination (TRT 0) to better protect their offspring from early PRRSV challenge. The metrics to answer the first question, the question of vaccine efficacy, involved a challenge study in 2-week old piglets with the main assessments being viremia and lung pathology. To answer the second question, the question of vaccine immunogenicity, we included a detailed analysis of the vaccine induced humoral and CMI response in gilts and piglets. 

Based on our main goal, this study did not include non-vaccinated gilts and non-challenged piglets. Yet, we can anticipate the following results within these groups based on the literature: (i) gilt vaccination can result in the transfer of maternal immune cells [8,25] and IFN-γ secreting cells; (ii) anti-PRRSV maternal-derived antibodies in unchallenged piglets would have remained over the course of the four-week study [26,27]; (iii) PRRSV-2 infection in piglets from unvaccinated gilts would have been more severe than in our treatments due to lack of maternal-derived antibodies and CMI transfer [13,27,28]. 

Vaccine efficacy analysis between our treatment groups clearly showed that our maternal AIV boost vaccination strategy (Figure 1) was able to at least partially protect the offspring; TRT 1 significantly decreased viremia and improved the piglet weight gain towards the end of the study; in addition, both TRT 1+2 significantly decreased lung pathology (Figure 2). This is in line with a previous study on maternal boost vaccinations using either MLV or AIV against a PRRSV-1 strain. While both booster vaccinations increased the farm-specific NA titers in the gilts, only the AIV provided the highest NA titers to the offspring and was detectable for five weeks after birth. However, both vaccinated groups had lower viremia upon PRRSV exposure in the nursery than offspring from unvaccinated gilts [27]. Combined, these data demonstrate that maternal AIV boost vaccinations can represent an efficient tool to limit the effect of both PRRSV-1 and 2 on pre-weaning piglets.

Vaccine immunogenicity analysis showed a clear effect of the AIV vaccinations on the humoral immune response in both gilts and piglets (Figure 3A,B). While MLV vaccination alone (TRT 0) did not induce NAs against the 1-7-4 challenge strain, both AIV boost vaccinations (TRT 1+2) induced considerable NA titers in most (TRT 2) or even all (TRT 1) gilts. Of note, with a correlation coefficient of r = 0.886 and *p* < 0.0001 these NA were strongly transferred to their offspring (Figure 3C). This strong correlation between maternal and offspring NA titers at weaning is consistent with previous research [26,29]. In most of these piglets, NA could be detected in serum until 4 wpi; at that time some of the TRT 0 piglets developed their own NA. As mentioned above, this timeline supports previous studies showing that maternal derived antibodies can stay in piglets for 4 weeks [26,27]. 

Combined with the vaccine efficacy data, these NA data demonstrate three main conclusions: (i) maternal AIV vaccination can efficiently boost the NA levels against their homologous PRRSV strain not only in the gilts but also in their offspring; (ii) these NA levels stay high in piglets until or even after weaning age (~ 2–4 wpi); and (iii) our results indicate that this AIV-boost induced NAs at least partially protected these early weaners from challenge with the autogenous PRRSV strain at 2 weeks of age. These data support previous studies showing protection by NAs through two passive transfer experiments to be protective in pregnant sows [30] and partially protective in young, weaned pigs [31]. Yet, follow-on studies have examined the difficulty of achieving NAs through vaccinations that provide subsequent challenge protection against homologous and heterologous PRRSV strains in nursery pigs [32,33,34,35,36] and in gilts and sows [37,38,39,40]. These studies are often strain and age dependent but in general they reveal the following: the generation of serum NAs is delayed against PRRSV and titers are lower than in other common swine disease vaccinations, and cross-reactivity against heterologous PRRSV strains is limited. Based on these limitations, it seems crucial to use an optimized maternal vaccination regimen such as an MLV prime/AIV boost combination to provide the best protection for piglets.

Compared to the humoral immune response, vaccine immunogenicity analysis of the CMI response showed a much lower difference between the treatment groups. This can be explained in one of two ways: either AIV did induce NAs but not a CMI response against the 1-7-4 strain, or the MLV-induced CMI response alone was already cross-reactive and the AIV boost effect was limited. While we cannot rule out either of these two options, our own (partially yet unpublished) results and the literature support the latter option: while NA titers often react only to homologous PRRSV strains, the CMI response exhibits greater cross-reactivity between heterologous PRRSV strains [13], combined with yet unpublished results showing limited cross-reactivity between heterologous PRRSV strains, and Ref. [32]. 

The most striking difference in the CMI response between the treatment groups was that at 0 wpi, pigs in TRT 2 had a higher systemic IFN-γ response in lymphocytes (Figure 4) and this increase was based on a stronger IFN-γ response in B cells and TCR-γδ T cells (Appendix A). At 2 wpi, B cells still had a higher IFN-γ response by number in the TrBr LN and significantly in the lung, and while TCR-γδ T cells did not show a response in the lung, by number their IFN-γ response was also higher in TrBr LNs (Figure 6). These data indicate that AIV vaccination not only has the potential to boost a NA response, but it can also boost the CMI response via IFN-γ producing B cells and TCR-γδ T cells. 

Different types of IFN-γ producing B cells have been reported in mice. “B effector 1” cells produced IFN-γ in response to IL-12 and IL-18 stimulation with this response being dependent on the IFN-γ receptor and the T-box transcription factor T-bet [41]. In addition, Bao et al. reported an inducible CD11a^hi^FcγRIII^hi^ B-cell population that they called “innate B cells”; in contrast to conventional B cells, upon CD40 ligation these innate B cells produced high amounts of IFN-γ [42]. Rodríguez-Carreño et al. studied IFN-γ producing cells by flow cytometry [43]. They did not find IFN-γ producing B cells in 6-month and 3-year old pigs after stimulation with phorbol myristate acetate (PMA)/ ionomycin. This discrepancy can be mainly explained by the very low frequency of IFN-γ producing B cells in older pigs (as shown for our gilts) and the data acquisition used by Rodríguez-Carreño et al.: in our gilts, <0.1% of B cells produced IFN-γ (d.n.s), while Rodríguez-Carreño et al. did not state the number of recorded cells, the figure allows one to estimate that the recorded B cell number is <1,000 with a background of ~30 cells in the IFN-γ^+^ B-cell gate. Hence, IFN-γ production by B cells at a rate of <0.1% would be reflected at less than one IFN-γ^+^ B-cell in their system and therefore, it could have not been detected in their system (Figure 2a of [43]). While IFN-γ producing B cells have not been described yet in pigs, Ogawa et al. showed in 2016 that compared to colostrum-deprived piglets, colostrum-fed piglets had significantly higher blood B cell numbers [7]. This demonstrates the successful transfer of B cells from sows to their offspring. Our data showing that B cells both in blood and TrBr LNs produce IFN-γ provide an indication that these B cells not only contribute to future antibody production, but they could also induce isotype class-switching from IgM to IgG [44]. Taken together, these data suggest an important role of IFN-γ producing B cells in the protection of piglets. Future studies on maternal transfer of immunity from sows to piglets should enclose a detailed analysis of B cells including these IFN-γ producing B cells.

Regarding the role of TCR-γδ T cells in early protection of piglets, Bandrick et al. showed in 2014 that TCR-γδ T cells are the main T-cell subset in sow colostrum and these cells also showed the strongest increase in piglets between pre- and after suckling [8]. This shows that these TCR-γδ T cells are highly transferred between gilt and piglet and together with our data, both studies support a role of TCR-γδ T cells in the protection of piglets. 

While IFN-γ production by B-cells and TCR-γδ T-cells was the only difference between the treatment groups, the detailed analysis of the CMI response revealed additional very interesting patterns of the anti-PRRSV response within the different locations and age groups. In lung tissue and BAL, CD8 T cells were consistently the strongest IFN-γ producers at necropsy, which was performed 2 and 4 wpi, so at 4 and 6 weeks of age. This is in line with our previous report in which we showed that CD8 T cells were the main lymphocyte subset in the lungs of PRRSV vaccinated or infected pigs at 9 wpi, or 13 weeks of age [13]. In addition to CD8 T cells, CD4 T cells also contributed to the local IFN-γ production in lung tissue and BAL. While the expression of the homing marker CCR7 on both of these T cell subsets from lymphoid and peripheral tissue was unsurprising (mixed in blood, Figure 5; mainly CCR7^+^ in TrBrLN and mainly CCR7^−^ in lung and BAL, Figure 7), the combined analysis of CD8α and CCR7 in CD4 T cells from lung and BAL revealed an unexpected result: the presence of a novel CD4 T-cell subset in pigs—CD8α^−^CCR7^−^ CD4 T cells (Figure 7). 

As mentioned above, CD8α and CCR7 are used in pigs to define three CD4 T cell subsets of increasing differentiation: CD8α^−^CCR7^+^ naïve, CD8α^−^CCR7^+^ TCM and CD8α^−^CCR7^+^ TEM CD4 T cells. The question is, at what stage of differentiation do these newly described CD8α^−^CCR7^−^ CD4 T cells fall in? CCR7 is used in several species, but while CD8α is the gold standard differentiation marker in porcine T cells, CD45RA is used in humans to differentiate CD45RA^+^ naïve from CD45RA^−^ memory cells. In 1999, Sallusto et al. studiedCD45RA and CCR7 expression in human CD4 and CD8 T cells in peripheral blood [45]. Based on this CD45RA and CCR7 expression, they could identify three CD4 subsets—one naïve CD45RA^+^CCR7^+^, and two memory—subsets, CD45RA^−^CCR7^+^ and CD45RA^−^CCR7^−^. They nicely demonstrated that while the CD45RA^−^CCR7^+^ cells expressed the lymph node homing marker, only the CD45RA^−^CCR7^−^ cells had an immediate effector function and therefore, they named these cells T_CM_ and T_EM_ cells, respectively. While Sallusto did not find CD45RA^+^CCR7^−^ in CD4 T cells, they found them in CD8 T cells. So, these CD45RA^+^CCR7^−^ CD8 cells represent the CD8 human counterpart to our newly described CD8α^−^CCR7^−^ CD4 T cells in pigs. These human CD45RA^+^CCR7^−^ CD8 T cells had a particularly prominent expression of the CD8 effector molecule perforin and this corresponded to a population of terminally differentiated CD27^−^ effector T cells [46]. These effector memory T cells re-expressing CD45RA were later termed T_EMRA_ cells [47]. In the meantime, these T_EMRA_ cells have also been described for CD4 T cells but their frequency is much lower than in human CD8 T cells and their role is less clear. These CD4 T_EMRA_ cells are more frequent in patients with some viral infections, such as dengue virus or cytomegalovirus, and they show a decreased CD27 expression but an increased expression of effector molecules [48]. Therefore, CD4 T_EMRA_ cells seem to have a similar terminal differentiation and strong effector function as their CD8 counterparts. Based on these human studies, we hypothesize that these newly identified porcine CD8α^−^CCR7^−^ CD4 T cells represent strong effectors and terminally differentiated CD4 T_EMRA_ cells.

In contrast to BAL and lung tissue, in blood and secondary lymphoid tissue, B cells were the main producers of IFN-γ until weaning age (0 and 2 wpi); their ongoing IFN-γ production further justifies research into IFN-γ producing B cells in young pigs. After weaning, so at 4 wpi, while B cells were still the main IFN-γ producers in the TrBr LNs, CD4 T cells became the major source of IFN-γ in blood. This trend seems to continue. In the older gilts, CD4 T cells were by far the strongest IFN-γ producers, producing ~60% of the total IFN- γ. The important role of these CD4 T cells in the systemic response against both PRRSV-1 and 2 has been repeatedly shown for post-weaning animals [13,49,50]. 

As so often in research, while trying to solve the questions on if, and how, maternal AIV boost vaccinations can improve protection against PRRSV challenge in piglets, we shed light on an even more complex immune response than originally thought. Thereby, we simultaneously raised new questions, not only on the role of CD8α^−^CCR7^−^ CD4 T cells, but especially on IFN-γ producing B cells: (i) “In addition to TCR-γδ T cells, why are mainly IFN-γ producing B cells transferred from mother to offspring?”; (ii) “How are these cells stimulated and which antigens do they recognize?”; (iii) “Are they rather innate B cells or B effector 1 cells?”; (iv) “What exact role do these IFN-γ producing B cells play in the protection of young pigs?”; (v) “What happens to these cells as the pig ages?”. So, it seems that the orchestra playing the awe-inspiring symphony/“harmonious complexity” of the maternal-derived immunity just got a little bigger [1].

## 5. Conclusions

This study provides five main take home messages for the reader: (1) maternal AIV boost vaccinations can improve the protection of pre-weaning piglets against the farm-prevalent PRRSV strain; (2) this protection is most likely based on (a) the transfer of neutralizing antibodies and (b) IFN-γ producing B cells; (3) these IFN-γ producing B cells are a major source of the systemic IFN-γ response in the early life of a piglet, until the standard weaning age; (4) post-weaning, CD4 T cells are the major contributors of systemic IFN-γ; and (5) while CD8 T cells are the major responders against PRRSV in lung and BAL, CD4 T cells also contribute to this response, including a novel CD8α^−^CCR7^−^ CD4 T-cell subset potentially representing porcine terminally differentiated CD4 T_EMRA_ cells. 

## Figures and Tables

**Figure 1 vaccines-09-00106-f001:**
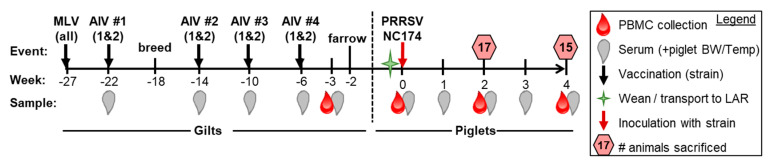
Study design: The study included gilts that were either vaccinated with a modified live vaccine only (MLV, treatment (TRT) 0), or an MLV boosted by a series of four vaccinations using one of two autogenous inactivated virus (AIV) vaccines (TRT 1+2). One week before farrowing, blood from gilts was collected to assess the systemic humoral and cell-mediated adaptive immune response. Piglets from three gilts with high humoral and T-cell responses within each treatment group (TRT 0, 1, 2) were included in the second part of the study—the challenge part at the North Carolina (NC) State Laboratory Animal Resources (LAR). These piglets were intranasally inoculated with the autogenous PRRSV 1-7-4 strain at two weeks of age (0 weeks post inoculation, wpi). Pigs were also bled, as illustrated, to assess the immune response to Porcine Reproductive and Respiratory Syndrome Virus (PRRSV). To assess lung pathology and study the local immune response, half of the pigs were sacrificed at 2 wpi and half at 4 wpi. PBMC: peripheral blood mononuclear cells.

**Figure 2 vaccines-09-00106-f002:**
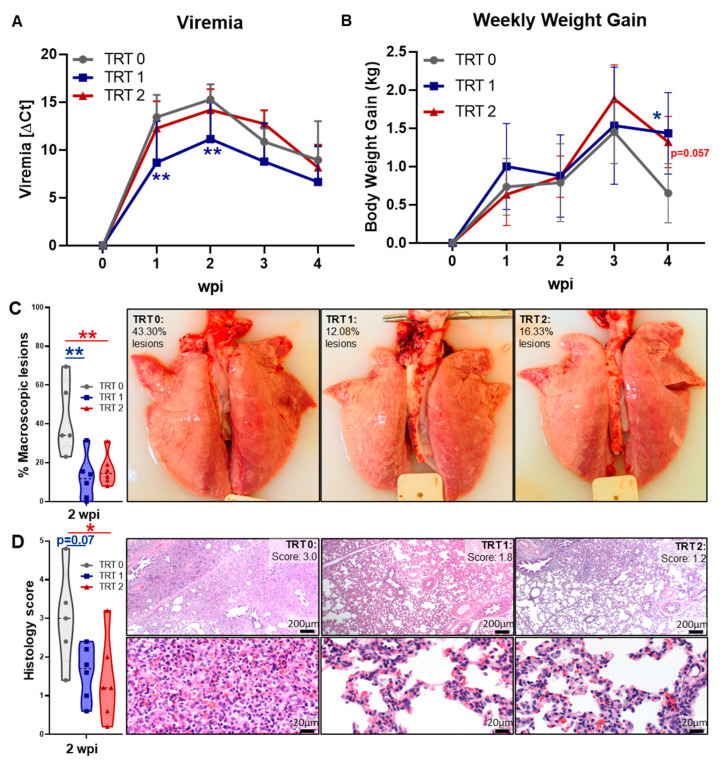
Maternal AIV vaccination boosts immunity to PRRSV-2 in 2-week old piglets: Viremia (**A**) and weekly body weight gains (**B**) were analyzed at different weeks post inoculation (wpi). For both, data were analyzed by mixed-effects model (REML) with Dunnett’s multiple comparisons test. (**C**) Percentage of macroscopic lung lesions, and (**D**) lung interstitial pneumonia histology scores at 2 weeks post inoculation. Data in (**C**) and (**D**) were analyzed by one-way ANOVA with Dunnett’s multiple comparison. All differences are shown within time point and compared to the TRT 0 control group – ** *p* < 0.01 and * *p* < 0.05.

**Figure 3 vaccines-09-00106-f003:**
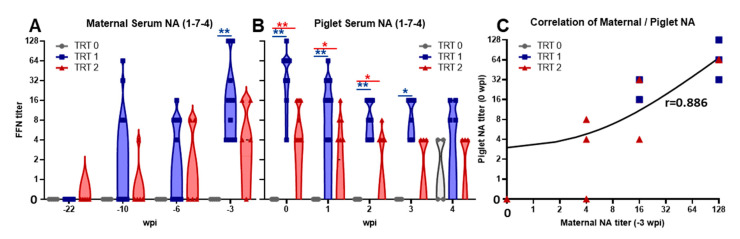
Autogenous inactivated boost vaccinations induce homologous neutralizing antibodies that are transferred to piglets. (**A**) Maternal serum neutralizing antibody (NA) titers to the homologous 1-7-4 strain after vaccination. (**B**) Piglet serum NA titers to 1-7-4 prior to inoculation (0 weeks post inoculation, wpi) and weekly after inoculation. (**C**) Pearson correlation matrix for maternal and piglet NA titers to 1-7-4 at -3 and 0 wpi, respectively. Data were analyzed via repeated-measures two-way ANOVA with Dunnett’s multiple comparisons. Significant differences are shown between treatments and within time point; they are designated as ** *p* < 0.01 and * *p* < 0.05.

**Figure 4 vaccines-09-00106-f004:**
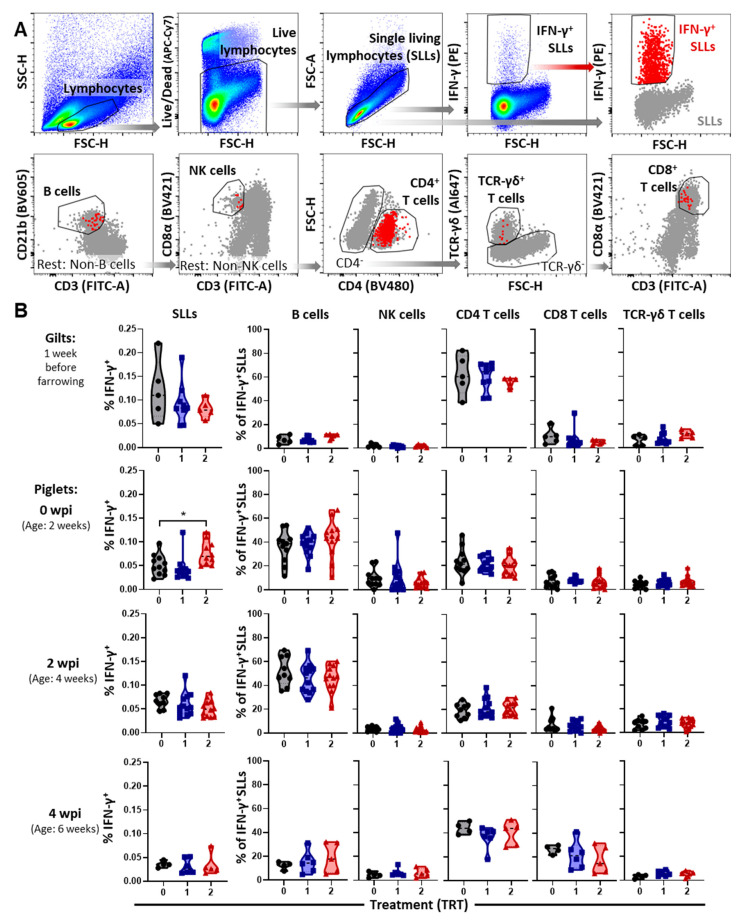
The systemic IFN-γ response to PRRSV 1-7-4 in gilts and their offspring: (**A**) shows the gating hierarchy starting with a forward/ side scatter (FSC/SSC) gate on lymphocytes, the exclusion of dead cells and doublets to focus the analysis of IFN-γ production on single living lymphocytes (SLLs). Subgates on B cells, NK cells, CD4, CD8 and TCR-γδ T cells were drawn on the whole SLL population (in gray) and overlaid by IFN-γ producing cells (in red). (**B**) shows the IFN-γ production as a percentage of all SLLs (left column) and the percentage of each immune cell subsets contributing to this IFN-γ production (all other columns). Data were analyzed via one-way ANOVA with Dunnett’s multiple comparisons. Significant differences between treatments are designated as * *p* < 0.05.

**Figure 5 vaccines-09-00106-f005:**
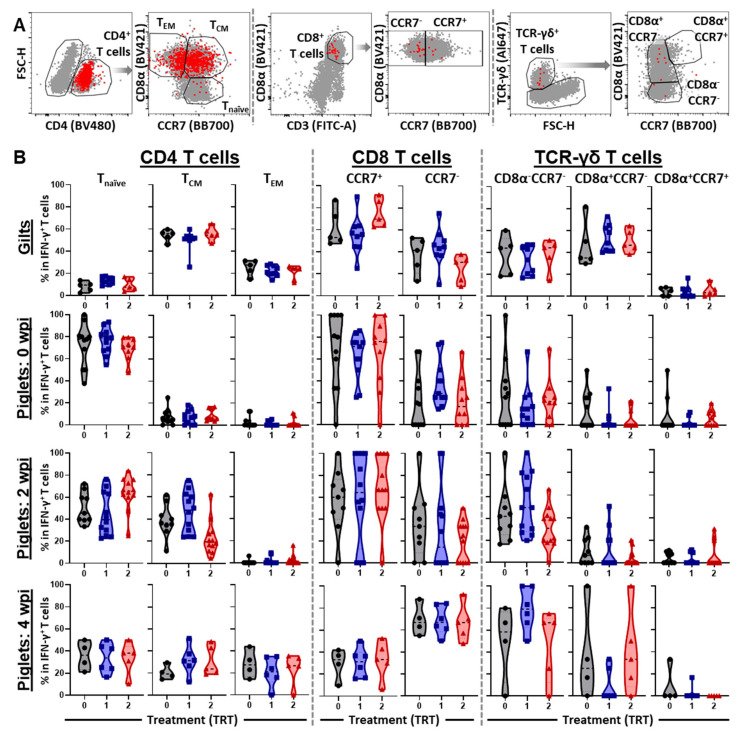
Differentiation of IFN-γ^+^ T-cell subsets in blood: (**A**) shows the gating used to differentiate CD4 naïve (T_naïve_, CD8α^−^CCR7^+^), central memory (T_CM_, CD8α^+^CCR7^+^), and effector memory (T_EM_, CD8α^+^CCR7^−^) T cells; it also shows the distinction of CCR7^+^ lymph node homing and CCR7^−^ peripheral tissue homing CD8 T cells. Additionally, it illustrates the different TCR-γδ T-cell subsets—naïve CD8α^−^CCR7^−^, and antigen experienced CD8α^+^CCR7^−^ and CD8α^+^CCR7^+^ cells. (**B**) shows the frequencies of these differentiation and homing defined subsets within IFN-γ producing CD4 T cells (left), CD8 T cells (middle), and TCR-γδ T cells (right) in gilts at one week before farrowing (upper row), piglets at 0 wpi (2nd row), 2 wpi (3rd row) and 4 wpi (4th row). Data were analyzed via one-way ANOVA with Dunnett’s multiple comparisons. No significant differences (*p* < 0.05) between groups were observed.

**Figure 6 vaccines-09-00106-f006:**
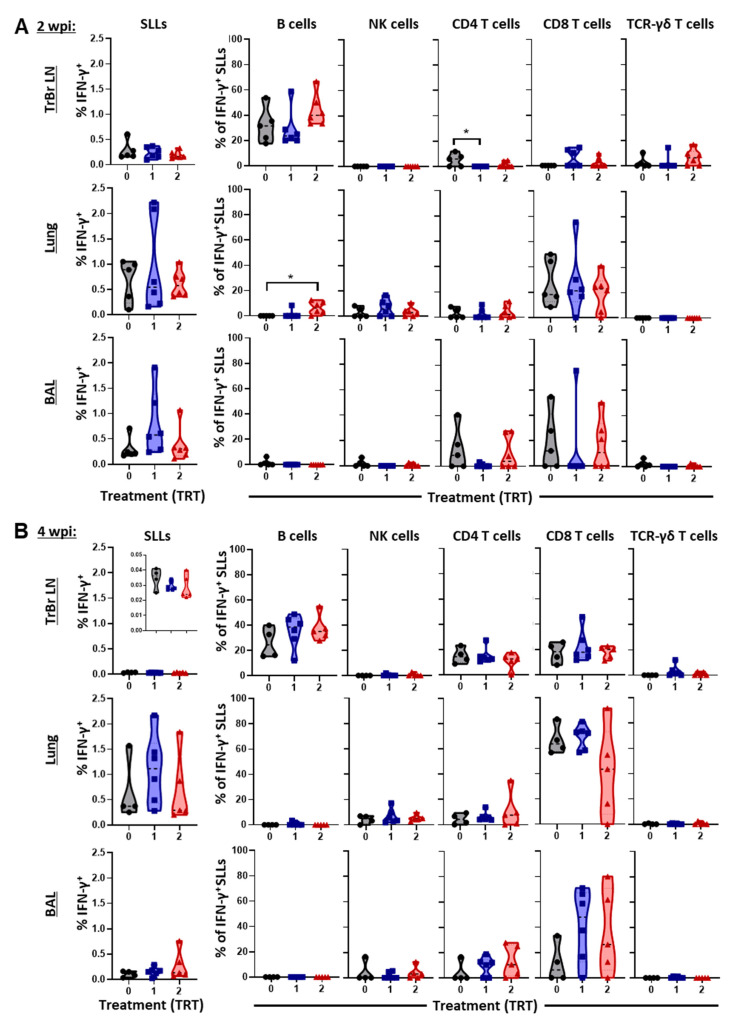
The local IFN-γ response to PRRSV 1-7-4 in bronchoalveolar lavage, lungs, and tracheobronchial lymph nodes of piglets. IFN-γ production as a percentage of all SLLs (left column) and the percentage of each immune cell subsets contributing to this IFN-γ production (all other columns) is shown within tracheobronchial lymph nodes (TrBr LN), lung tissue, and bronchoalveolar lavage (BAL) at 2 wpi (**A**) and 4 wpi (**B**). Data were analyzed via one-way ANOVA with Dunnett’s multiple comparisons. Significant differences between treatments are designated as * *p* < 0.05.

**Figure 7 vaccines-09-00106-f007:**
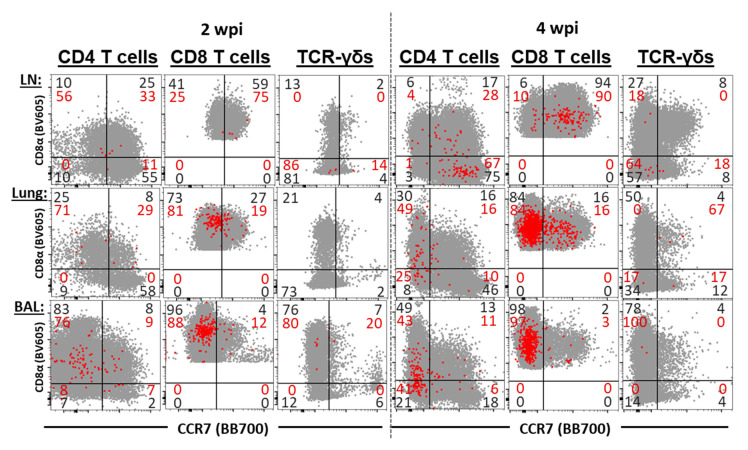
Differentiation of IFN-γ^+^ T-cell subsets in tracheobronchial lymph nodes, lung tissue and bronchoalveolar lavage (BAL): These overlays show the CCR7 (x-axis) and CD8α (y-axis) expression of CD4, CD8, and TCR-γδ T-cell subsets (gray) and the respective IFN-γ^+^ T-cell subsets (red) from all animals in tracheobronchial lymph nodes (LN), lung tissue, and BAL at 2 wpi (three left columns) and 4 wpi (three right columns). No statistical analysis was performed.

**Table 1 vaccines-09-00106-t001:** Flow cytometry antibody staining panel.

Antigen	Clone	Isotype	Fluorochrome	Labeling Strategy	Primary Ab Source	2nd Ab Source
CD3	PPT3	IgG1	FITC	Directly conjugated	Southern Biotech	-
CD4	74-12-4	IgG2b	Brilliant Violet 480	Secondary antibody	BEI Resources	Jackson Immunoresearch
CD8α	76-2-11	IgG2a	Brilliant Violet 421	Secondary antibody	BEI Resources	Jackson Immunoresearch
CD21α	BB6-11C9.6	IgG1	Brilliant Violet 605	Biotin-streptavidin	Novus Bio	Biolegend
TCR-γδ	PGBL22A	IgG1	Alexa Fluor 647	Directly conjugated	Kingfisher	Invitrogen
CCR7	3D12	rIgG2a	Brilliant Blue 700	Directly conjugated	BD Biosciences	-
IFN-γ	P2G10	IgG1	PE	Directly conjugated	BD Biosciences	-
Live/Dead	-	-	Near Infra-red	-	Invitrogen	-

Note: FITC = Fluorescein isothiocyanate; BEI = Biodefense and Emerging Infections.

## Data Availability

The data presented in this study are available on request from the corresponding author.

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
