# Peer review of "Maternal Autogenous Inactivated Virus Vaccination Boosts Immunity to PRRSV in Piglets"

_vaccines, 2021, doi:10.3390/vaccines9020106_

Round 1

Reviewer 1 Report

Many abbreviations are not determine din the first time used - e.g. CCR7, NA, NC174 (fig 1), FCM, MOI and CASY (section 2.6), TEMRA cells (line 100, these introduced later in the results, but not here)- please correct throughout the manuscript (see section 3.5 in https://www.mdpi.com/authors/layout#_bookmark14)

The immune system of most domestic species neonates is developed at 2-4m of age - why you only covered age up to 6 weeks while the piglet will need protection until its immune system is fully developed?

As the thymus will be the origin of B cell production, how would you differ between self-produced B cells and the maternal B-cells?

Introduction

Is TRT acceptable shortcut? (I assume it is for 'treatment'/'treatment group')

It is not clear why you chosen to follow IFN-gamma in your experiment design?

(A short reasoning of this rational will help the reader)

Aims are presented in the start of the methods section (the convention is usually in the end of the introduction section; several discussion points are 'hidden' within the result section. Consider changing this structure/flow issues.

Methods

Lines 143-144, necropsy is not a cause of death... reword this sentence.

Line 150 - company of the PBS?

Section 2.3 - suggest to reword/shorten the description and put the service lab names in brackets.

Reference to deltaCT method is missing.

Section 2.5 - company producing the Euthasol?

Some of the description is too long - e.g. no need to say who preformed routine H&E stains...

Line 195 - % of formalin used?

Section 2.6 - are there citations for the two software mentioned in Lines 215-217?

Line 218 - which figure shows which tissue?

Line 224 - Version and company of REML + citation/web-link?

Line 228 - Citation for the statistical test needed.

Results -

Fig 2 A & B - Without reading the article text, it is not clear from the figure which comparison is significant here. For example, is TR1 significant in comparison to to both TR0 and TR2 results (A panel)?

D- How you determined 'pneumonia histology scores'? Is there a range there?

Lines 247-253 - The relationship between the score to the % of lesions withing the lungs is not clear - are these comparable?

There are a lot of non-significant result sin figures 4,5, and 6 - is it necessary to show these?

Figure 7 - add 'LN' as abbreviation of 'Lymph node' withing the legend text.

Why 'no statistical analysis was performed'?

Discussion -

Line 429 - add year after 'Osorio'; Line 498 - add year after 'Sallusto et al.'

Line 5o9 - reference is missing (for Temra CD4 cells)

Suppl Figure 2 - what '0' '1' and '2' represents?

Author Response

Vaccines-1042388: Revision 1 – Response to reviewer 1:

General comments:

Many abbreviations are not determined in the first time used - e.g. CCR7, NA, NC174 (fig 1), FCM, MOI and CASY (section 2.6), TEMRA cells (line 100, these introduced later in the results, but not here)- please correct throughout the manuscript (see section 3.5 in https://www.mdpi.com/authors/layout#_bookmark14)

Response: We now introduced CCR7, NA, FCM and MOI. NC174 is not an abbreviation but a strain name. CASY is the name of an instrument and TEMRA is also more a term than an abbreviation as explained in line 510: “…these effector memory T cells re-expressing CD45RA were later termed TEMRA cells.”

The immune system of most domestic species neonates is developed at 2-4m of age - why you only covered age up to 6 weeks while the piglet will need protection until its immune system is fully developed?

Response: We agree that the immune system of pigs mature over time: It will increase its antigen repertoire; it will develop more antigen-experienced and memory cells and thereby become more responsive. Yet, maternal immunity if most effective during the early stages of life and especially before weaning. Hence, we focused our analysis on the effect of maternal immunization on protecting pre-weaning piglets from PRRSV challenge. We then followed the infection and immune response until 2 and 4 weeks post infection; these time frames correspond to the peak of lung pathology and the anti-PRRSV T-cell response, respectively.

As the thymus will be the origin of B cell production, how would you differ between self-produced B cells and the maternal B-cells?

Response: The bone marrow is the origin of B- and T-cell production. The thymus is involved in the development and selection of T cells. Due to lack of access to markers for differentiation of maternal and offspring B cells in the pig model, we did not include such markers.

Introduction

Is TRT acceptable shortcut? (I assume it is for 'treatment'/'treatment group')

Response: We now introduced “treatment” in line 91.

It is not clear why you chosen to follow IFN-gamma in your experiment design?

(A short reasoning of this rational will help the reader)

Response: IFN-gamma is a crucial cytokine in anti-viral responses and a standard measure to assess the cell-mediated response against PRRSV. We included a statement highlighting the role of IFN-g in lines 83-85.

Aims are presented in the start of the methods section (the convention is usually in the end of the introduction section; several discussion points are 'hidden' within the result section. Consider changing this structure/flow issues.

Response: We state the aims at the end of the introduction in lines 91-102 (starting with “Therefore, the main goal of this study…”).

Methods

Lines 143-144, necropsy is not a cause of death... reword this sentence.

Response: Done.

Line 150 - company of the PBS?

Response: Done.

Section 2.3 - suggest to reword/shorten the description and put the service lab names in brackets.

Response: This section is less than 7 lines, so we do not find any way to shorten this description. We corrected the description of the method to “reverse transcription” qPCR (RT qPCR).

Reference to deltaCT method is missing.

Response: The calculation is included in lines 182-184.

Section 2.5 - company producing the Euthasol?

Response: Included in line 192/3

Some of the description is too long - e.g. no need to say who preformed routine H&E stains...

 Response: We deleted the exemplified text.

Line 195 - % of formalin used?

Response: The fixative is a prepared, ready-to-use solution purchased from EMS. This company does not provide additional information on the contents, probably for proprietary reasons.

Section 2.6 - are there citations for the two software mentioned in Lines 215-217?

Response: The CytExpert is the flow cytometry data acquisition software provided by the manufacturer of the flow cytometer (Beckman Coulter). The FlowJo software is the gold standard for flow cytometry analysis. Both are highly recognized programs in the field; hence, we do not feel the need to include citations here.

Line 218 - which figure shows which tissue?

Response: Lines 218-220 state: “Gating hierarchy for the analysis of the IFN-γ response is shown in Figures 4 and 5 for PBMC and in Supplementary Figure 1 and Figure 7 for BAL, lung tissue, and TrBr LNs.” Accordingly, Figures 4 and 5 show PBMC; Supplementary Figure 1 and Figure 7 show all other tissues – BAL, lung tissue, and TrBr LNs.

Line 224 - Version and company of REML + citation/web-link?

Response: REML is a mathematical method for fitting linear mixed models. We included the full title of this method in line 226

Line 228 - Citation for the statistical test needed.

Response: Based on the nature of the data and the required analyses, Graphpad Prism recommends specific statistical analyses. We followed the recommendations of the program. Since the statistical tests need to be adapted to the specific dataset and analysis, we do not see how a citation would further help with the analysis of our specific dataset and comparisons.

Results

Fig 2 A & B - Without reading the article text, it is not clear from the figure which comparison is significant here. For example, is TR1 significant in comparison to to both TR0 and TR2 results (A panel)?

Response: We included a more specific statement in the figure legend to better point out these comparisons.

D- How you determined 'pneumonia histology scores'? Is there a range there?

Response: The pneumonia scoring follows a previously described standard procedure with the reference included in line 201 of the materials and methods section.

Lines 247-253 - The relationship between the score to the % of lesions withing the lungs is not clear - are these comparable?

Response: We did not perform a correlation analysis between these two parameters. Yet, nearly all animals from the TRT 1+2 groups are below the TRT0 group in both parameters; this indicates that both parameters have similar trends.

There are a lot of non-significant result sin figures 4,5, and 6 - is it necessary to show these?

Response: The relevance of these figures lays in describing the immune response to PRRSV in young piglets which has not been performed in such detail yet. Therefore, while there are no between group differences, these figures are still highly relevant for the scientific community to improve the understanding of the anti-PRRSV response in young pigs.

Figure 7 - add 'LN' as abbreviation of 'Lymph node' withing the legend text.

Response: The abbreviation is included in this figure legend in line 380.

Why 'no statistical analysis was performed'?

Response: As stated in the figure legend, the figure shows overlays of all animals to better visualize the responding T-cell subsets. Based on the low frequency of IFN-gamma producing T cells, we preferred to only perform only a descriptive analysis to prevent potential overinterpretation of the data.

Discussion

Line 429 - add year after 'Osorio'; Line 498 - add year after 'Sallusto et al.'

Response: Since we reference several publications in the sentence of line 429, a single year is not possible; hence, we deleted the inclusion of the Osorio lab. We included the year for the Sallusto reference.

Line 5o9 - reference is missing (for Temra CD4 cells)

 Response: This reference (now reference 47) is included in line 511.

Suppl Figure 2 - what '0' '1' and '2' represents?

Response: As indicated in the axis description, these numbers represent the different treatments (TRT).

Reviewer 2 Report

Comments to the Authors of manuscript number: vaccines-1042388 entitled “Maternal autogenous inactivated virus vaccination boosts the piglet humoral and cell-mediated

immunity to PRRSV via transfer of neutralizing antibodies and interferon-gamma producing B cells”.

Authors have presented the influence of maternal vaccination on humoral and cell-mediated immunity of their offspring. This study is very interesting, however it should correct before the acceptance:

  1. L 73 small letter
  2. L 77- 78 this sentence should be rewritten
  3. L 118 “ high responding” – what is the meaning – it should be explained
  4. L 134, 180, 185, – Authors should uniform used term. Gilts is young female which is no mother, thus gilts or sows?
  5. L144 small letter
  6. L 142-143 not clear sentence
  7. L190 was injection i.m. or i.v.?
  8. L196 there is a lack one sentence about common histology

9.L 248, 250, 256 – lung pathology? It should be indicated what …

  1. L 251 – these lesions were observed but compared to which group?
  2. L 254 – decrease in lung pathology – it should be explained
  3. L268, 317,297,347, 396, 414, 424  gilts?
  4. L 326 results in some parts looks like the discussion. It should be corrected
  5. L 406-411 too long sentence
  6. L 425 in general weaning is performed at the age of 4 weeks. If we added 2-3 wpi we have piglets at the age of 7 weeks. Were in this study piglets at this age? Because there is not clear what the meaning “young piglets” How long piglets are young in the mind of Authors?
  7. L 431 gestational gilts/sows-? It is not clear

17.L 439, 442, 534,  – small letter

  1. L 461 – what is the meaning of “older piglets” in the accordance to breeding
  2. L 462 – gilts

19.L 428, 473. 530 – “young piglets” the same as above

20.L 480 – “very early stages of life” – it should be explained

Author Response

General comments:

Comments to the Authors of manuscript number: vaccines-1042388 entitled “Maternal autogenous inactivated virus vaccination boosts the piglet humoral and cell-mediated

immunity to PRRSV via transfer of neutralizing antibodies and interferon-gamma producing B cells”.

Authors have presented the influence of maternal vaccination on humoral and cell-mediated immunity of their offspring. This study is very interesting, however it should correct before the acceptance. 

Itemized comments:

L 73 small letter

Response: Done.

L 77- 78 this sentence should be rewritten

Response: Done.

L 118 “ high responding” – what is the meaning – it should be explained

Response: We changed this statement to “with high humoral and T-cell responses”

L 134, 180, 185, – Authors should uniform used term. Gilts is young female which is no mother, thus gilts or sows?

Response: We started with gilts that became sows once the piglets were born; to avoid confusion, we now uniformly use “gilt”.

L144 small letter

Response: Done.

L 142-143 not clear sentence

Response: We simplified the sentence to “In summary, there were a total of 36 piglets in the study with 12 piglets per treatment group.”

L190 was injection i.m. or i.v.?

Response: i.v. ; this information is now included in the manuscript.

L196 there is a lack one sentence about common histology

Response: We do not know which information are missing. Since reviewer 1 suggested to decrease the information provided in this section, we decided to keep the text as is with the exception to not name the histology core as specified by reviewer 1.

L 248, 250, 256 – lung pathology? It should be indicated what …

AND L 254 – decrease in lung pathology – it should be explained

Response: % macroscopic lesions for macroscopic pathology and interstitial pneumonia for histopathology – now included in lines 253/4 and 256.

L 251 – these lesions were observed but compared to which group?

Response: This is compared to the control treatment group TRT 0; this information is now included in line 257.

L268, 317,297,347, 396, 414, 424  gilts?

Response: Yes. See above.

L 326 results in some parts looks like the discussion. It should be corrected

Response: Since some of the information required to follow the results section is rather novel, we decided to include an introductory sentence in some result paragraphs; this shall better orient the reader before s/he is confronted with the rather complex staining patterns. Since these sentences do not discuss our results and to facilitate the readability of the results section, we would prefer to keep these introductory sentences.

L 406-411 too long sentence

Response: We cut this sentence into two sentences.

L 425 in general weaning is performed at the age of 4 weeks. If we added 2-3 wpi we have piglets at the age of 7 weeks. Were in this study piglets at this age? Because there is not clear what the meaning “young piglets” How long piglets are young in the mind of Authors?

Response: To avoid confusion, we deleted “young” and only kept “piglets until or even after weaning age”. We also corrected the weaning age to the adequate “~2-4 weeks”.

L 431 gestational gilts/sows-? It is not clear

Response: To avoid confusion, we deleted “gestational” and changed this statement to “gilts and sows”.

L 439, 442, 534,  – small letter

Response: Done

L 461 – what is the meaning of “older piglets” in the accordance to breeding

Response: A timeline for the kinetic of IFN-gamma producing cells in pigs is not available. So we cannot state at what age the decrease occurs. Our intention was to contrast the frequency of IFN-gamma producing B cells in our “younger” piglets from the “older” gilts. Since we are not able to provide a specific age for our “older” statement, we included a statement that refers to our gilts to better clarify this statement.

L 462 – gilts

Response: Done

L 428, 473. 530 – “young piglets” the same as above

Response: As above, since these statements are adequate for piglets of all ages, we deleted the “young” to avoid confusion.

L 480 – “very early stages of life” – it should be explained

Response: As above, since there is no timeline for their role available in pigs and it is highly likely to be a fluent process, we cannot provide specific ages. To avoid confusion, we deleted this statement.

Reviewer 3 Report

Dear Author,

Submitted manuscript vaccines-1042388 “Maternal autogenous inactivated virus vaccination boosts the piglet humoral and cell-mediated immunity to PRRSV via transfer of neutralizing antibodies and interferon-gamma producing B cells” is a nice, well-planned work and drafted concisely. The work shows a thorough study in which a very exhaustive discussion of the literature has been carried out. Introduction provides sufficient background, and the other sections clearly presented and analyzed exhaustively.

The Authors have investigated an interesting topic related to piglet immunity. The subject is adequate with the overall journal scope.  The present form of manuscript is well written, presented and discussed, and understandable to a specialist readership. Here are some minor comments which authors can take into consideration.

Title of the manuscript is too long. In place of humoral and cell-mediated immunity, “immune response” and only “B Cells” is sufficient.

Remove the background color from all the graphs if you want to use box-violin graph-type. Background color is hiding the population number and distribution.

AIV abbreviated as autogenous inactivated vaccines; at several places “AIV vaccines” have been used.

Scoring strategy of macroscopic lesions and histology score need to be mention little bit here as well. Table-1 details of the flow cytometry antibodies can go into supplementary information.

Figure 1 needs formatting as unnecessary line numbers is there. I am expecting that author will take care of other formatting errors and English language errors at their end.

In general, the organization and the structure of the article are satisfactory and in agreement with the journal instructions for authors.

All the very Best.

Author Response

General comments:

Submitted manuscript vaccines-1042388 “Maternal autogenous inactivated virus vaccination boosts the piglet humoral and cell-mediated immunity to PRRSV via transfer of neutralizing antibodies and interferon-gamma producing B cells” is a nice, well-planned work and drafted concisely. The work shows a thorough study in which a very exhaustive discussion of the literature has been carried out. Introduction provides sufficient background, and the other sections clearly presented and analyzed exhaustively.

The Authors have investigated an interesting topic related to piglet immunity. The subject is adequate with the overall journal scope.  The present form of manuscript is well written, presented and discussed, and understandable to a specialist readership. Here are some minor comments which authors can take into consideration.

Itemized comments:

Title of the manuscript is too long. In place of humoral and cell-mediated immunity, “immune response” and only “B Cells” is sufficient.

Response: We shortened the title to “Maternal autogenous inactivated virus vaccination boosts immunity to PRRSV in piglets.”

Remove the background color from all the graphs if you want to use box-violin graph-type. Background color is hiding the population number and distribution.

Response: We assume this issue occurs on a print-out since we do not see this issue on screen. We discussed the graph type with a number of colleagues and tried to balance on the one side clarity and on the other side specificity. Without the background color, the individual data points might be better visible but the figure itself becomes too inordinate. Therefore, we would prefer to keep the light background color.

AIV abbreviated as autogenous inactivated vaccines; at several places “AIV vaccines” have been used.

Response: Done.

Scoring strategy of macroscopic lesions and histology score need to be mention little bit here as well.

Response: We included the requested scoring strategy in the text – now at lines 194 – 204.

Table-1 details of the flow cytometry antibodies can go into supplementary information.

Response: We agree that this table could go into the supplementary information. Yet, this would leave the main manuscript without any information on the used antibodies/ staining strategy. Based on the complexity and importance of the staining, we therefore prefer to keep it in the methods section to facilitate that the reader has all relevant information in the main manuscript. In addition, we searched the literature for other rather complex flow panels and most manuscripts that used a table to provide similarly complex antibody panel data had it in the main text.

Figure 1 needs formatting as unnecessary line numbers is there.

Response: Done.

Round 2

Reviewer 2 Report

Comments to the Authors of manuscript number: vaccines-1042388 entitled “Maternal autogenous inactivated virus vaccination 2boosts immunity to PRRSV in piglets”.

Authors have presented the influence of maternal vaccination on humoral and cell-mediated immunity of their offspring. It is visible that Authors have corrected the manuscript, however it should further correct before the acceptance:

  1. L 114 “ high response” – what is the meaning – it should be explained- it is considered as the concentration or level of some cytokines, factor?

When is low response and when is high? What is the limit?

Is it possible to distinguish low and high response based on high-titer, high-avidity antibody response?

It should be explained. What concentration and what factor is considered as high response from? Potency of humoral immunity by enhancing key elements of the B-cell respons..

Reference?

  1. In the figure is written sows, but in the text gilts. There is substantial difference between gilts and sows. All the text should be corrected properly.

A gilt is a young female pig. In common use, gilt is used to refer to a pig that has not yet been bred, whether only a few months old or approaching a year. Technically, however, the term gilt is defined as a female pig that is less than six months old. A gilt is intact, or capable of breeding and producing young, and her reproductive organs are not surgically or chemically altered.

Author Response

Dear Editor and Reviewer 2,

Please find enclosed the revised version of the manuscript vaccines-1042388 with the new title “Maternal autogenous inactivated virus vaccination boosts immunity to PRRSV in piglets”.
Thank you very much for your critical but constructive comments. We have again included your suggestions in the text using the “Track changes” function to highlight the changes. Below, we have responded to both items: i) We included a detailed explanation of the selection criteria and process; and ii) we changed the missed “sows” to gilts and included a statement explaining the use of “gilts” vs. “sows” in the manuscript to avoid any confusion. We hope that these modifications satisfy this reviewer and the editor to accept this manuscript.
The manuscript has not been submitted for publication in another journal or published elsewhere; and it has been approved by all co-authors. Please feel free to contact me if you require any additional information.
We would highly appreciate the publication of this manuscript in the journal Vaccines.

Thank you and yours faithfully,

Tobias Kaeser

General comments:
Authors have presented the influence of maternal vaccination on humoral and cell-mediated immunity of their
offspring. It is visible that Authors have corrected the manuscript, however it should further correct before the
acceptance:
Itemized comments:
1. L 114 “ high response” – what is the meaning – it should be explained- it is considered as the concentration or
level of some cytokines, factor? When is low response and when is high? What is the limit? Is it possible to
distinguish low and high response based on high-titer, high-avidity antibody response? It should be explained.
What concentration and what factor is considered as high response from? Potency of humoral immunity by
enhancing key elements of the B-cell response. Reference?
Response: We provided a more detailed explanation of the selection criteria and process in the revised manuscript – lines
132-139: “…, we selected three gilts per group to provide the piglets for the challenge study. The selected gilts had an
adequate litter size (>7), and within their group the highest serum fluorescent focus neutralization (FFN) titer (primary
immune criteria) and the highest CD4 T-cell IFN-γ response (secondary immune criteria) against the AIV challenge strain.
For TRT 1+2, these FFN titers ranged from 1:4 to 1:128. Since TRT 0 did not have any titers against the AIV strain, we chose
the gilts with the highest anti-MLV titers; these titers ranged from 1:16 to 1:32. The IFN-γ response within all selected gilts
ranged from 0.09-0.21% of total CD4 T cells. Each of these gilts provided four piglets for the challenge study.” Since the
selection criteria were a high response within the groups/animals of this study, a reference is not available. To further
avoid confusion, we deleted the statement “high responder” gilts as well (Line 138).
2. In the figure is written sows, but in the text gilts. There is substantial difference between gilts and sows. All the
text should be corrected properly. A gilt is a young female pig. In common use, gilt is used to refer to a pig that
has not yet been bred, whether only a few months old or approaching a year. Technically, however, the term gilt
is defined as a female pig that is less than six months old. A gilt is intact, or capable of breeding and producing
young, and her reproductive organs are not surgically or chemically altered.
Response: As the reviewer indicates, the technical terminology differs from the common use of gilts vs sows. To avoid
any confusion, we added a clarification statement for the use of gilts vs. sows in this manuscript: Lines 123-125 “To
maintain consistent terminology throughout this study, the enrolled sexually mature female pigs (here defined as gilt)
will be referred to as gilts even after they were bred and gave birth (commonly referred to as sows).” In addition, we
corrected the terminology in Figure 1 from “Sows” to “Gilts”.

Round 3

Reviewer 2 Report

Dear Authors,

Thank you very much for your patience, but the rule is that not only the author has to understand the text he wrote. If there is a commonly accepted nomenclature, it sounds a bit strange:"Piglets from three gilts with high humoral". because:

  1. how gilts can have offspring..
  2. how we can define - high

Now, the understanding of the text is better.